# Knowledge Storage and Extraction in Language Models (Part A)[*]

## Abstract

Large language models can store extensive world knowledge, often *extractable* through question-answering (e.g., "What is Abraham Lincoln's birthday?"). However, it's unclear whether the model answers questions based on exposure to exact/similar questions during training, or if it genuinely extracts knowledge from the source (e.g., Wikipedia biographies). In this paper, we conduct an in-depth study of this problem using a controlled set of semi-synthetic biography data. We uncover a relationship between the model's knowledge extraction ability and different *diversity measures* of the training data. We conduct (nearly) linear probing, revealing a strong correlation between this relationship and whether the model (nearly) linearly encodes the knowledge attributes at the hidden embedding of the entity names, or across the embeddings of other tokens in the training text.

## 1    Introduction

Knowledge is crucial for human cognition and communication, allowing us to comprehend and utilize information. For humans, this often involves memorization, the process of storing and retrieving information in the brain. For example, after reading a biography of Abraham Lincoln, we can memorize the information and later answer questions like "Where was Lincoln born?" or "What is Lincoln's birthday?" Memorization enables us to extract and manipulate knowledge from the sentences we read or hear, recognize the entities, relations, and facts expressed in the text, and apply logical and causal reasoning to infer new information or answer queries.

In this paper, we explore **how transformer based language models** memorize knowledge during training and extract it during inference. This is distinct from in-context learning or RAG (Lewis et al., 2020), where the model is given a paragraph during inference and immediately answers questions about it. We focus on *factual knowledge* (e.g., knowledge graph) that a language model needs to memorize from the training corpus, encode in its weights, and extract later during inference.

We stress that *memorizing* all sentences in the training data **does not** ensure that the model can *extract or manipulate* the factual knowledge from the sentences during inference. Language models can reproduce the exact input during inference, but this doesn't necessarily mean they can use these sentences to answer factual questions related to them. Hence, we differentiate between "memorization of knowledge" in language models and traditional memorization in machine learning, which merely means the model can fit the exact training data, but doesn't imply the model can **extract the knowledge flexibly** from the data after training.

For example, if the training data includes Lincoln's biography, the model can memorize and reproduce the sentence "Abraham Lincoln was born in Hodgenville, K.Y." when given the prompt "Abraham Lincoln was born in", but it might not be able to answer the question "Which city was Abraham Lincoln born in?" Therefore, a key question is:

> *How do language models memorize knowledge during training, and extract it later to answer questions or perform logical reasoning during inference?*

Previous works have demonstrated that language models can "memorize" a lot of knowledge by probing the model to answer questions related to different entities and attributes, see Omar et al.

---

[*]Since "knowledge" is a broad subject, we have to write separate papers to cover its different aspects. This Part A addresses how knowledge is *stored*, the conditions under which knowledge can be *extracted* through instruct fine-tuning, and introduces probing techniques. A natural subsequent question concerns how such knowledge can be further *manipulated* for downstream tasks. This is explored in our Part B (Anonymous, 2023). We've anonymously submitted both to ICLR 2024 as standalone papers, ensuring no result overlap and making each self-contained. Our Part B is also in the supplementary material for interested readers.

(2023); Singhal et al. (2022); Sun et al. (2023) and the citations therein. However, these studies use models trained on internet data, leaving it **unclear** whether the model answers questions like "Which city was Abraham Lincoln born in?" by *extracting knowledge* from Abraham Lincoln's biography (our focus) or if it encountered a similar, or even the same question during training and simply memorized the answer (traditional memorization).

Given the challenges of conducting controlled experiments with internet data, we propose studying this question using well-controlled, synthetically generated data,[1] examining the models' mathematical properties that characterize their knowledge representation and extraction. We construct a synthetic dataset of $100k$ biographies, including their birthday, birth city, major of study, etc. We also use LLaMA (Touvron et al., 2023) to rewrite them to make them close to real-life biography styles. We pretrain the language model on the biography dataset of all the $100k$ people. We ask:

*After training a language model on the biography dataset, can the model be finetuned to extract the knowledge to answer questions like "Where is the birth city of [name]" or "What did [name] study?", and if so, how does the model achieve so?*

We evaluate our model's knowledge extraction ability by finetuning it on question and answers (QAs) for a $p$ fraction of individuals and testing its ability to answer QAs about the remaining $1-p$ fraction. This training and testing process ensures that the model sees enough data to understand the QAs, and also isolates the effect of knowledge extraction from other factors like seeing the exact same question during training. The paper is structured as follows:

1. In Section 3, we demonstrate that training a model on all biographies and QAs for a $p$ fraction of individuals together in pretraining time enables it to (apply knowledge to) answer questions about the remaining $1-p$ fraction. We call this process *mixed training*. We also observe in mixed training, the model learns in an unconventional way: it *first uses QAs* to encode knowledge about the $p$ fraction, then correlates this encoded knowledge with the biography to infer generalization to the remaining $1-p$ fraction. This learning process deviates from typical human learning and is less frequently used in large language model training.

2. In Section 4, we examine a model pre-trained on biographies and fine-tuned on QAs for a $p$ fraction of individuals. It struggles to answer questions for the remaining $1-p$ fraction, *regardless of model size, pre-train time, and finetune parameters*. However, accuracy significantly improves with knowledge augmentations like varying writing styles or sentence permutations. Even if this augmentation is applied to a subset of individuals, what we call celebrities, test accuracy for others also increases significantly. The mere inclusion of celebrity data in pre-training enhances the model's knowledge extraction for minorities. One of our work's key contribution is **establishing this strong link** between knowledge augmentation in pre-training data and model's improved knowledge extraction after fine-tuning.

3. In Section 5, **as another main contribution**, we use (nearly) linear probing techniques to show that knowledge augmentation compels the model to encode a person's knowledge almost linearly in the model's hidden embedding of the person's name tokens. Without augmentation, the model encodes the person's knowledge across all biography words/tokens, making knowledge extraction during finetuning nearly impossible. We summarize this as:

   **no knowledge augmentation in data** $\iff$ **attribute is not entirely stored on person's names**
   $$\iff \textbf{knowledge cannot be extracted via instruct finetune}$$

4. In Appendix B, we show that BERT-like models, pre-trained on biography data and finetuned on QAs, cannot extract a person's knowledge after finetuning, regardless of the bio-data knowledge augmentation used during training, unless the knowledge is a single word or multiple but independent words (like birth month, day, and year).

**Related work.** LINEAR PROBING OF KNOWLEDGE. Linear probing is a recognized method to examine how a model encodes knowledge (Aspillaga et al., 2021; Conneau et al., 2018; Dai et al., 2021; Geva et al., 2020; Li et al., 2021; Meng et al., 2022; Sun et al., 2023). Contrary to previous studies that suggest models trained on internet data can linearly encode knowledge in the hidden

---

[1]One could suggest filtering the data to eliminate such questions and retraining the model. However, this doesn't rule out the presence of similar sentences "Which city did Abraham Lincoln grow up in?", more complex ones in French, or grammatically incorrect versions like "Where Abraham Lincoln birth in?" in the data.

embeddings of entity names, we find that such encoding is only possible with knowledge augmentations like permutation/rewriting of entity-attribute knowledge during pretraining. Without these augmentations, the language model can still memorize the training data, but it is not linearly encoded in the entity's hidden embeddings, making knowledge extraction via QAs quite hard, if not impossible, even with instruct fine-tuning. This implies that diverse internet data on the same entity is vital for pre-training the language model for knowledge extraction during inference. The usefulness of augmentations of pretraining data for language models was also empirically observed in literature Berglund et al. (2023); Cai et al. (2020); Eldan & Li (2023); Kobayashi (2018), but they did not explore **where** the knowledge is nearly-linearly encoded in a sentence and its correlation with knowledge augmentation, a process we refer to as P-probing in Section A.1.

PROBING LANGUAGE MODELS' KNOWLEDGE VIA QAS. Question answering (QA) is a common method to probe the knowledge encoded in language models pretrained on internet data (Hernandez et al., 2023; Naseem et al., 2021; Omar et al., 2023; Peng et al., 2022; Petroni et al., 2019; Richardson & Sabharwal, 2020; Singhal et al., 2022; Sun et al., 2023). However, it's unclear whether these models answer questions by extracting knowledge from the training source or by recognizing exact/similar questions from training. We use semi-synthetic data in a controlled experiment for out-of-distribution testing on individuals whose QAs were not part of training. This approach also allows us to study the correlation between knowledge extraction and the diversity of pretrain data.

ENCODER VERSUS DECODER FOR QAS. While BERT-based models Kenton & Toutanova (2019) are also used for knowledge extraction through QAs (Choi et al., 2022; Sushil et al., 2021), our work indicates that they are less effective at extracting knowledge compared to GPT models.

## 2 PRELIMINARIES

In this paper, we analyze synthetic human biography datasets and near-real datasets generated by LLaMa-30B (v1) (Touvron et al., 2023; Zhou et al., 2023). Detailed descriptions are in the appendix, with a brief overview here.

**BIO dataset bioS.** The synthetic dataset, bioS, generates profiles for $N = 100,000$ individuals. Each individual's details are randomly and *independently* selected from a uniform distribution. The birth dates offer $200 \times 12 \times 28$ possibilities, while other categories offer $100 \sim 1,000$ choices. We also add a "company city" attribute which *depends* on the employer's headquarters location. We ensure uniqueness in each individual's full name.

We generate a six-sentence biographical text entry for each individual, highlighting six distinct aspects. For diversity, each sentence is randomly chosen from approximately 50 distinct templates. In the basic configuration, we generate a single biographical entry for each person, maintaining a consistent order for the six sentences. We use "bioS single" to denote this basic configuration. See an example entry below:

Anya Briar Forger was born on October 2, 1996. She spent her early years in Princeton, NJ. She received mentorship and guidance from faculty members at Massachusetts Institute of Technology. She completed her education with a focus on Communications. She had a professional role at Meta Platforms. She was employed in Menlo Park, CA.

$$(2.1)$$

We also explore 3 types of knowledge augmentations: (1) multi$M$, generating $M$ biography entries for an individual using varied templates, (2) fullname, substituting he/she/they with the person's full name; and (3) permute, shuffling the six sentences randomly. Examples are given in Section 4.2.

**BIO dataset bioR.** We examine a "close-to-real" dataset produced by LLaMA-30B (Touvron et al., 2023; Zhou et al., 2023). For the set of $N = 100,000$ individuals, we provide an instructive prompt to LLaMA to generate a biographical entry. Here's an example:

Anya Briar Forger is a renowned social media strategist and community manager. She is currently working as a Marketing Manager at Meta Platforms. She completed her graduation from MIT with a degree in Communications. She was born on 2nd October 1996 in Princeton, NJ and was brought up in the same city. She later moved to Menlo Park in California to be a part of Facebook's team. She is an avid reader and loves traveling.

We diversified our instructive prompts by drawing from a pool of templates and employed rejection sampling to guarantee the inclusion of all six attributes. In the basic configuration, we produce a single biographical entry for each person (denoted as "bioR single"). For comparison, we also consider multi$M$ augmentation which generates $M$ entries per person and the fullname augmentation. Additional examples can be found in Appendix C.

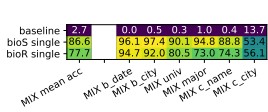

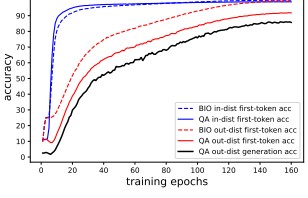

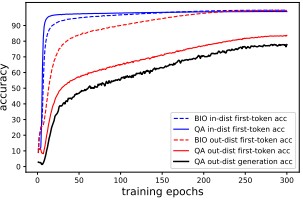

(a) QA out-dist accuracies     (b) training on bioS dataset     (c) training on bioR dataset

Figure 1: Accuracies and loss curves for mix training. b_date,b_city,c_name,c_city stand for birth date, birth city, company name, company city, and mean acc stands for the mean accuracy of the six attributes. Baseline is majority-guessing (c_city has large accuracy because many companies are based in NYC).

**QA dataset.** This paper explores the effectiveness of a trained language model in retaining knowledge from BIO data. As discussed in the introduction, memorization *is more than just predicting the next token* when given exact sentences from BIO. It includes the model's ability to truly **extract knowledge from the BIO**. We assess this knowledge extraction using a question and answer (QA) framework. For each individual, we pose six questions targeting their six unique attributes:

1. What is the birth date of Anya Briar Forger?
   Answer: October 2, 1996.
2. What is the birth city of Anya Briar Forger?
   Answer: Princeton, NJ.
3. Which university did Anya Briar Forger study?
   Answer: Massachusetts Institute of Technology.
4. What major did Anya Briar Forger study?
   Answer: Communications.
5. Which company did Anya Briar Forger work for?
   Answer: Meta Platforms.
6. Where did Anya Briar Forger work?
   Answer: Menlo Park, CA.

For each question, we use it as a prompt for the model to generate a response. QA accuracy is measured by the proportion of answers that match the correct response.

**Model architectures.** The standard GPT2-small architecture comprises 12 layers with 12 heads and 768 dimensions Radford et al. (2019). Due to GPT2's limitations from its absolute positional embedding, we use its modern rotary positional embedding variant Black et al. (2022); Su et al. (2021), referred to as GPT2 for brevity. We retain the GPT2 small architecture (124M) for pre-training on the bioS data, but use a larger 12-layer, 20-head, 1280-dim GPT (302M) for the bioR data to accommodate its increased complexity. The default GPT2 tokenizer is used, which converts simple words into single tokens, but names and most other attributes into tokens of varying lengths.[2]

**Training.** We investigate two types of autoregressive training, detailed in Appendix D.

PRETRAIN + INSTRUCT FINETUNE. Here, we pre-train the language model on the BIO data, randomly sampling and concatenating them into 512-token sentences, separated by a standard <|EOS|> token. The model is then fine-tuned using half of the QA data and evaluated on the remaining half, mirroring the typical instruct finetune process.

MIX TRAINING. In mix training, we pre-train the model on all BIO data and half of the QA data. BIO and QA entries are randomly sampled without requiring them to be from the same individual. We use a parameter $QA_r$ to control the QA data amount, primarily setting $QA_r = 0.8$ (a $2:8$ BIO to QA entry ratio). The model's generation accuracy is evaluated using the remaining QA data.[3]

**LoRA finetune.** In full finetuning a pretrained model is tuned for a downstream task such as QAs. LoRA finetuning (Hu et al., 2021) improves upon this by freezing all pretrained model parameters and adding low-rank updates to a subset of the weight matrices for fine-tuning. We apply a low-rank update to the query/value matrices of the transformer model and the embedding layer to account for input data distribution shifts. Full finetuning is also included when presenting negative results.

## 3 MIX TRAINING

Mix training involves training the model using BIO data for *all* individuals and QAs for half of them. The group of individuals whose QAs are included in the training set is referred to as *in-distribution* or $\mathcal{P}_{\text{train}}$. The model's generative accuracy is then tested on the QAs from the remaining individuals ($\mathcal{P}_{\text{test}}$) to assess its out-of-distribution generalization capability.

As shown in Figure 1(a), a mix-trained model exhibits strong out-of-distribution generalization, answering most QAs with mean accuracies of $86.6\%$ for bioS and $77.7\%$ for bioR. This indicates

---

[2]Only in Figure 2 when presenting a negative result, we tried a 12-layer, 32-head, 2048-dim GPT (682M).
[3]See Appendix E for a comparison of how $QA_r$ affects performance.

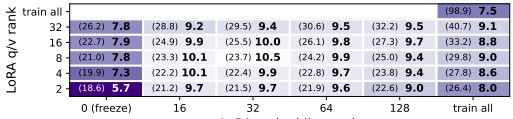

(a) 124M model, pre-trained 540 passes on bioS

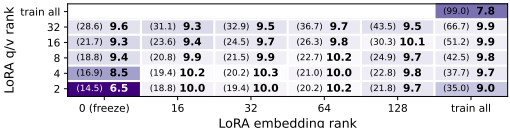

(b) 302M model, pre-trained 1000 passes on bioR

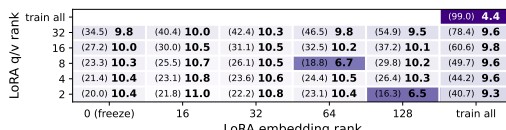

(c) 682M model, pre-trained 1350 passes on bioS

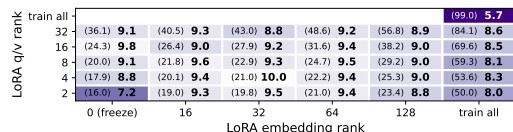

(d) 682M model, pre-trained 1350 passes on bioR

Figure 2: BIO pretrain + QA finetune (train acc) / **test acc**. Bold number indicates QA generation accuracy on $\mathcal{P}_{\text{test}}$, and the smaller number in bracket represents QA (first-token) accuracy on $\mathcal{P}_{\text{train}}$. For LoRA fine-tune we consider a rank $r = 2, 4, 8, 16, 32$ update on the query/value (q/v) matrices and a rank $r' = 0, 16, 32, 64, 128$ update on the word embedding matrix. More details are in Appendix F.

that the model can extract and utilize knowledge from the BIO data, addressing queries about an individual's attributes even when no QA about that person was used in training; only their BIO entry was provided. However, our detailed analysis reveals that the model employs a somewhat unconventional method to extract knowledge through mix training.

### 3.1 MODEL'S ABNORMAL LEARNING BEHAVIOR

We examine the model's mixed training process for knowledge storage and extraction by monitoring its accuracies on the BIO/QA data and for $\mathcal{P}_{\text{train}}/\mathcal{P}_{\text{test}}$ separately. Specifically,[4]

- BIO first-token accuracy: we track the model's next-token-prediction accuracy on the first token of each of the six attributes (birthdate, birthcity, etc.) in the BIO data, separately for $\mathcal{P}_{\text{train}}/\mathcal{P}_{\text{test}}$. This measures the model's BIO data memorization performance. (Despite all individuals' BIO data appearing in training, we still separately track them for $\mathcal{P}_{\text{train}}/\mathcal{P}_{\text{test}}$.)
- QA first-token accuracy: we track the model's next-token-prediction accuracy on the first answer token in the QA data, separately for $\mathcal{P}_{\text{train}}/\mathcal{P}_{\text{test}}$. This loosely estimates the model's QA generation performance.
- QA generation accuracy: we track the model's whole-attribute generation accuracy on $\mathcal{P}_{\text{test}}$.

From Figure 1(b) and 1(c), we find that the model employs an unconventional learning strategy.

- Initially, the model uses the QA data from the training set to encode knowledge for people in $\mathcal{P}_{\text{train}}$, as indicated by the rapid increase in QA in-dist accuracy. This also aids in memorizing in-dist BIO data, as shown by the subsequent rise of the BIO in-dist accuracy.
- The model then gradually aligns the encoded knowledge with the BIO data to learn to extract knowledge and generalize it to $\mathcal{P}_{\text{test}}$. Notably, it takes a while before the BIO out-dist accuracy catches up, followed by an increase in the QA out-dist accuracy.

This is akin to the "study to pass the test" approach in schools, where students prepare using past exam questions and textbooks for answers. While this may yield high scores, it doesn't reflect the natural progression of human knowledge acquisition. **To address this**, we explore a scenario in the next section where the model is pretrained on the BIO data without exposure to the questions. [5]

## 4 BIO PRETRAIN + QA INSTRUCT FINETUNE

We now examine a scenario where the model is pre-trained solely on the BIO data of all individuals. It is then fine-tuned using QAs from half of these individuals, denoted as $\mathcal{P}_{\text{train}}$, without further use

---

[4]Interested readers may consider "whole-attribute" accuracies instead of "first-token" accuracies. They are similar, so we omit them here.

[5]In mixed training, we selected $\text{QA}_r = 0.8$, maintaining a $8 : 2$ QA to BIO ratio as outlined in Section 2. We found a higher QA ratio improves QA test accuracy (Figure 10 in Appendix E), further supporting our observation of the model's abnormal behavior: it first learns knowledge from QA and then associates it with BIO. For comparison, LLaMA was trained using only 2% of tokens from StackExchange (Touvron et al., 2023).

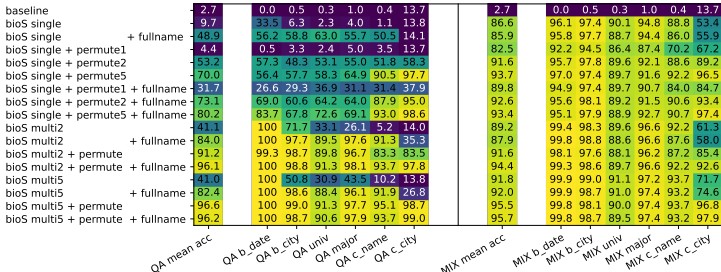

Figure 3: Comparison of BIO pretraining + QA finetuning (left) versus their mixed training counterparts (right) under various knowledge augmentations on the bioS data. Displayed values indicate QA generation accuracies for six attributes in $\mathcal{P}_{\text{test}}$. Refer to Figure 12 for bioR data and Appendix F for more details. **Observation.** Knowledge augmentation in pretraining data improves model generalization to out-of-distribution QAs after finetuning. Accuracy increases with more augmentations introduced; while mixed training is minimally impacted by knowledge augmentation.

of biographies. The model's generalization is evaluated on questions related to the remaining half, denoted as $\mathcal{P}_{\text{test}}$, whose BIO/QA data were not used during fine-tuning. This process mirrors human knowledge acquisition, where learning from textbooks is applied to later answer exam questions.

## 4.1 MODEL MAY FAIL TO EXTRACT KNOWLEDGE AFTER PRETRAINING ON BIO DATA

We first pretrain on the basic bioS and bioR datasets, each containing a single biography per person. The QA finetune generalization accuracies (on $\mathcal{P}_{\text{test}}$) are reported in Figure 2, using both full and LoRA finetuning (Hu et al., 2021). The model's QA finetune training accuracy on $\mathcal{P}_{\text{train}}$ is also included for comparison.

Despite a 99+% first-token accuracy during pretraining (see Appendix E), the model exhibits near-zero QA accuracy on $\mathcal{P}_{\text{test}}$ for all finetuning parameters. This suggests that while the model can memorize the BIO data token-by-token, it struggles to extract the underlying knowledge. Full-finetuning achieves high *in-distribution* QA finetune accuracy (nearly perfect on $\mathcal{P}_{\text{train}}$), indicating it can memorize the QAs for individuals in the finetuning set. However, it is largely ineffective for QAs concerning individuals in $\mathcal{P}_{\text{test}}$. In sum, we observe:

perfect BIO token memorization + perfect QA answers for half the people

$\not\Longrightarrow$ correct QA answers for the other half.  (*knowledge extraction does not come for free*)

This holds true even when the model size is approximately 7000 times larger than $N = 100k$, the number of individuals, each individual is observed 1350 times during pretraining, and numerous finetune parameters have been explored. Despite memorizing all knowledge from the BIO data during pretraining, the model encodes it in a disorganized manner within the transformer, preventing knowledge extraction during finetuning.[6]

Figure 2 seems to contradict the success of large models like GPT3.5, trained on internet data such as Common Crawl and known for effective knowledge extraction upon fine-tuning. Why is this? Analyzing the test accuracy breakdown for the six attributes on the bioS data (Figure 3, the "bioS single" row), we see that QA fine-tuning in fact achieves a 33% generalization accuracy on the "birthdate" attribute but fares poorly on others. This is because our bioS single data consistently places birthdate as the first attribute after a person's name, unlike internet data which presents information variably, often repeating it with diverse wordings and orderings.

## 4.2 KNOWLEDGE AUGMENTATION

We explore how knowledge augmentation enhances a model's capacity to store and efficiently retrieve knowledge from training data. We focus on three augmentations: adding multiplicity, introducing permutations, and repeating full names, typically found in internet data. The original datasets without augmentation are referred to as bioS single and bioR single.

---

[6]This is not a result of catastrophic forgetting, a common issue during heavy fine-tuning where the model forgets the pretraining data. Even with LoRA fine-tuning, which introduces minimal low-rank updates to model weights while preserving the pretrained model, test accuracy only slightly improves.

- MULTIPLICITY. We denote the method of creating $M$ distinct biography entries for each individual, using varied language but retaining the same information, as $\mathsf{multi}M$.[7] An example of adding multiplicity to the biography in (2.1) is:

   *Anya Briar Forger* came into this world on *October 2, 1996*. She originated from *Princeton, NJ*. She pursued advanced coursework at *Massachusetts Institute of Technology*. She dedicated her studies to *Software Engineering*. She developed her career at *Meta Platforms*. She gained work experience in *Menlo Park, CA*.

- PERMUTATION. We denote adding random permutations to the biography sentences as $\mathsf{permute}$.[8] For instance, the example above can be permuted as follows:

   *Anya Briar Forger* originated from *Princeton, NJ*. She dedicated her studies to *Communications*. She gained work experience in *Menlo Park, CA*. She developed her career at *Meta Platforms*. She came into this world on *October 2, 1996*. She pursued advanced coursework at *Massachusetts Institute of Technology*.

- FULLNAME. We denote the augmentation where all pronouns or partial names in $\mathsf{bioS}/\mathsf{bioR}$ are replaced with the person's full name as $\mathsf{fullname}$.

**Results.** In Figure 3, we present our results for the $\mathsf{bioS}$ dataset. (Parallel results for the $\mathsf{bioR}$ dataset are in Figure 12.) We implemented each knowledge augmentation individually and in combinations, then compared the model's QA finetune accuracy on $\mathcal{P}_{\text{test}}$ using LoRA. The model architecture and training parameters remained consistent, but the pre-training datasets varied based on the applied augmentations. Further details are in Appendix F.

We find that adding multiplicity, permutations, or repeating full names all improve the model's ability to memorize the person's information during pretraining, making knowledge extraction easier later.[9] Notably, pretraining on a dataset where each individual has five diverse biography entries (i.e., different wording, different sentence shuffling) boosts the QA fine-tune accuracy (on $\mathcal{P}_{\text{test}}$) from 9.7% to 96.6%. Moreover, such accuracy increases as data multiplicity or permutation number increases, highlighting the model's improved ability to store and extract knowledge when presented with repeated information during pretraining.

One might infer that exposing the model to varied expressions of identical knowledge encourages it to focus on the underlying logical structure of the information, rather than its superficial presentation. This could foster a more direct link between an individual's name and their attributes. We will introduce probing techniques to substantiate this hypothesis in Section 5.

### 4.3 CELEBRITY CAN HELP MINORITY

The previous subsection highlighted the significant benefits of knowledge augmentation. However, in practice, we may not have augmented data for all individuals. This subsection explores whether partially augmenting data can improve knowledge extraction for non-augmented data. In our biography dataset, the augmented subset is akin to a "celebrity" group with plentiful online biographical information, potentially included in the fine-tuning dataset as well. The non-augmented subset is comparable to a "minority" group with limited biographical data.

For comparison, we introduce an additional set of $N = 100,000$ individuals, the celebrity group $\mathcal{P}_{\text{cel}}$, while the original $N$ individuals form the minority group $\mathcal{P}_{\text{min}}$. We test both synthetic $\mathsf{bioS}$ and more realistic $\mathsf{bioR}$ data. For $\mathsf{bioS}$, the celebrity group's biographies use the $\mathsf{multi5+permute}$ augmentation, simulating varied expressions found on internet. For $\mathsf{bioR}$, the celebrity group uses the $\mathsf{multi5}$ augmentation, generating their biographies five times using LLaMA.

The language model is pretrained on the combined set $\mathcal{P}_{\text{cel}} \cup \mathcal{P}_{\text{min}}$ biographies and then fine-tuned using QAs from the celebrity group $\mathcal{P}_{\text{cel}}$. We evaluate the model's QA accuracy on the $\mathcal{P}_{\text{min}}$ group.[10] Our results are presented in Figure 4.

---

[7]For $\mathsf{bioS}$ data, each of the six sentences is selected from around 50 templates, with a new template resampled for each sentence in the $M$ entries. For $\mathsf{bioR}$ data, we recreate the biography using LLaMA for each of the $M$ entries.

[8]For $\mathsf{bioS\ single}$, we denote random permutation of the same six sentences $P$ times as $\mathsf{permute}P$. For $\mathsf{bioS\ multi}M$, we denote random permutation of each of the $M$ biography entries as $\mathsf{permute}$. The $\mathsf{bioR}$ data, generated by LLaMA, already has some randomness in sentence ordering, so no extra permutations are added.

[9]An exception is when permutation is directly added to the single data without multiplicity (see "$\mathsf{bioS\ single + permute1}$"), this hurts the QA performance as it makes knowledge extraction harder.

[10]Other fine-tuning variations, such as QA fine-tuning with half of $\mathcal{P}_{\text{min}}$ as training and half as testing, show negligible differences.

| | QA mean acc | | QA b_date | QA b_city | QA univ | QA major | QA c_name | QA c_city |
|---|---|---|---|---|---|---|---|---|
| baseline | 2.7 | | 0.0 | 0.5 | 0.3 | 1.0 | 0.4 | 13.7 |
| bioS single + permute1 | 4.4 | | 0.5 | 3.3 | 2.4 | 5.0 | 3.5 | 13.7 |
| bioS single + permute1 + CEL | 86.8 | | 98.3 | 96.8 | 90.7 | 90.2 | 71.7 | 80.1 |
| bioR single | 10.0 | | 25.1 | 13.9 | 2.4 | 5.5 | 2.0 | 14.1 |
| bioR single + wiki | 7.3 | | 18.4 | 5.2 | 2.6 | 4.3 | 1.8 | 14.1 |
| bioR single + CEL | 76.3 | | 94.3 | 85.3 | 82.9 | 79.4 | 67.0 | 56.6 |

Figure 4: QA finetune accuracy on the *minority group* with vs. without celebrity data in the pretraining process. Experiment details are in Appendix I, where we also include additional experiments in Figure 16.

**Results.** In the synthetic bioS case, introducing celebrity data boosts the minority group's QA accuracy from 4.4% to 86.8%. This is significant because:

- the minority group's BIO pretrain data *remains unchanged* in both cases, with $\mathcal{P}_{\min}$ using bioS single+permute1 for biographies, and

- the minority group's QA data *is not used* during fine-tuning.

This highlights that **simply including celebrity data during pretraining** significantly improves the model's ability to store and extract knowledge from the minority group. Similarly, in the more realistic bioR case, introducing celebrity data also increases the minority group's QA accuracy from 10.0% to 76.3%. We believe this strongly suggests that this phenomenon *also occurs in real-world scenarios*. We will introduce probing techniques to validate the above findings in Section 5.

*Remark* 4.1. Using the bioR dataset, we find the positive impact of celebrity data is *not universal*. Substituting it with the WikiBook dataset improves the model's English comprehension, yet it still struggles with biographical knowledge extraction. This suggests that only celebrity data of *similar form* truly aids knowledge extraction for minority groups. In Figure 16 in Appendix I, we further investigate different celebrity data types and instances of minor format differences between minority and celebrity knowledge.

## 5 Knowledge Probes on the BIO Pretrained Model

We investigate how a pretrained language model on BIO data encodes knowledge in its hidden representations using two probing techniques: position-based probing (P-probing) and query-based probing (Q-probing). Both techniques employ simple (nearly-linear) probes to extract a person's attributes from the model's hidden representations. Detailed findings are in Appendix A.

**In P-probing,** we input biography entries into the pretrained model and train a linear classifier on the last hidden layer to predict six target attributes. To accommodate varied data lengths, we identify six *special token positions* preceding the first occurrences of the six attributes in each biography entry. We use the transformer's last hidden layer at these positions to (linearly) predict the six target attributes (Figure 5).[11] Our results (Figure 6) show that *increased knowledge augmentation in the pretrain data improves P-probing prediction accuracies from earlier token positions*. In the basic bioS single setup, P-probing accuracy remains low until the token immediately preceding the target attribute. This suggests the model memorizes BIO data but encodes knowledge in a complex manner, revealing a person's attribute **only after encountering all prior attributes**. This **prevents knowledge extraction during QA finetuning**, particularly when only the person's name is given. In Appendix A, we use a Venn diagram to precisely illustrate which attribute is stored after observing another, further confirming this finding.

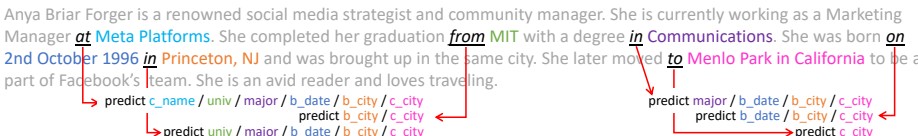

Figure 5: Illustration of the P-probing. Underscore prepositions are the *special token positions* where we prob. The task is to predict all attributes following these positions. Given the attribute ordering, there can be up to $6 \times 6 = 36$ tasks across all data.

---

[11] For each target attribute prediction task, we freeze the pretrained network but add a trainable rank-2 update on the embedding layer to account for the task change.

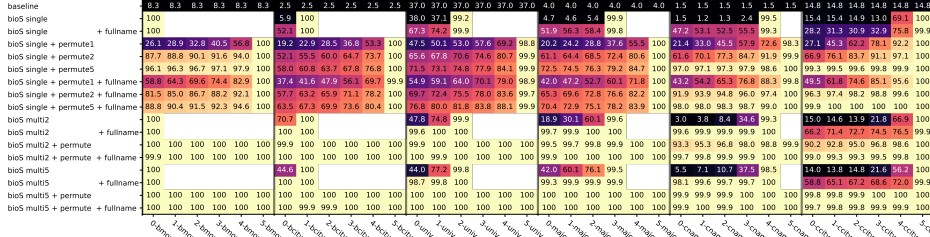

Figure 6: P-probing accuracies for various pretrained models on bioS data. Each **row** represents a pretrained model using a different knowledge augmentation, and each **column** labeled "$i$-$field$" shows the accuracy of predicting the *first token* of $field$ from position $i$. Details are in Section 5 and Appendix G (where we also include experiments for the bioR data and for predicting the full-attribute $field$.)

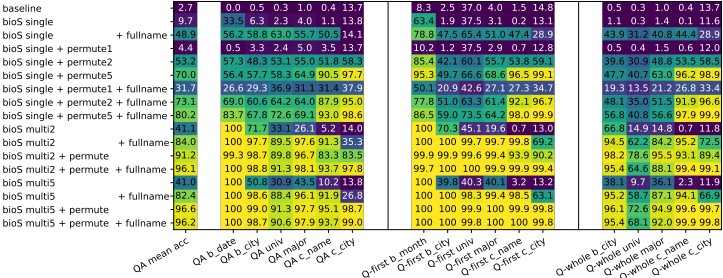

Figure 7: Q-probing accuracies. Each **row** denotes a pretrained model with its specific knowledge augmentation. The left block reiterates QA finetune accuracies from Figure 3. The middle showcases Q-probing accuracies on the first-token prediction for the six attributes, and the right focuses on Q-probing for the whole-attribute prediction. (Further details for bioR and more are in Appendix G. Note: For birth date, first token predicts the whole birth month; we do not have whole-attribute prediction for it since it has too many choices.)

**In Q-probing,** we focus on the knowledge directly linked to a person's name. We evaluate input sentences *containing only* the person's full name and train a linear classifier on the last layer's hidden states to predict the person's six attributes.[12] Our results (Figure 7 in Appendix A.2) show that the knowledge-extraction finetune accuracy *is directly linked to* whether the knowledge is (nearly-)linearly stored on the person's name in the pretrained model. This is a property independent of the finetune parameters, and suggests that the model *does not utilize contextual or global information from the biographies to extract knowledge about the individual.*

## 6 CONCLUSION

This study explores the ability of pre-trained language models to store and extract knowledge during inference using question-answering tasks. We created a semi-synthetic biography dataset and utilized probing techniques to examine the effect of knowledge augmentation on the storage and extractability of knowledge in pre-trained transformers. Synthetic data offers increased control over model training and fine-tuning inputs, which is crucial for understanding the influence of different data sources on the **internal mechanisms** of transformers. This could potentially be a significant future direction for unraveling the complexities of transformers. The paper also highlights the **importance of rewriting** essential but infrequently occurring data during pre-training to ensure its effective storage for subsequent tasks. This should be achieved using tools like ChatGPT before pre-training, as rectification during the fine-tuning stage might be too late if the pre-training data has not been fully augmented. While our primary focus was on autoregressive language models, our techniques are also applicable to bidirectional models like BERT, as discussed in Appendix J.

---

[12]We freeze all transformer layers (acquired through pretraining), except the embedding layer, to which we apply a rank-16 update. This adjustment is arguably the minimal change necessary since we are tackling a notably different input distribution.

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
