# OpenReview forum: "Knowledge Storage and Extraction in Language Models (Part A)"
_ICLR.cc/2024/Conference — ICLR 2024 Conference Withdrawn Submission_

### Official Review · Reviewer_HX8x · 2023-10-26

**Soundness:** 2 fair
**Presentation:** 2 fair
**Contribution:** 3 good
**Rating:** 3
**Confidence:** 5

**Summary:**

This paper explores how LMs capture the factual knowledge: by constructing an internal KG or simply memorized QA pairs.
To do that, the authors use LLaMA to construct a synthetic data set and train GPT-2 with it. Then, the verification is done by prompting and probing.

**Strengths:**

1. The topic is good. This paper studies a very interesting research problem. Figuring it out can help people better understand how LMs encode factual knowledge, and how to better enhance/edit factual knowledge.
2. The synthetic setting is novel. Researching the factual knowledge captured during pretraining is fairly intractable, since it's hard to annotate facts from the pretraining corpus and run analysis experiments on pretraining. This paper constructs a synthetic pretraining environment using LLaMA and reproduces the pretraining of small GPT-2.

**Weaknesses:**

My main concern is the experiment setting. The general design is okay, but some essential points are missing. I'll elaborate in detail below.
  1. The template to construct the synthetic dataset is problematic.

      a) First of all, it's not diverse. Even if using LLaMA, it works more like a simple verbalizer. We know that factual knowledge is sparse in the text (compared to linguistic knowledge): some facts exist across multiple sentences. But in the synthetic dataset, facts are verbalized in a quite limited way. For analysis, it's fine since LM is trained in a general way. However, this paper includes the pretraining part. I don't think pretraining with this dataset is reasonable. It's easy for LM to learn shortcuts.

      b) Second, the biography is too short and quite similar to each other. LMs may capture and extract facts relating to one person or topic. Such short and similar text might cause LMs to be confused when learning facts. Even for humans, it's hard to differentiate persons/topics with such short and similar introductions.

2. Contributions 1 and 2 are not clear to me.

      a) The synthetic data is generated randomly, and there could be conflicts between each other. Researching the generalization is a bit confusing (contribution 2).

      b) While for contribution 1, it seems the authors just try to test if GPT-2 generalizes well to understand the question and extract facts for answering. I don't know if GPT-2 is pretrained from scratch or not (see question 2). If yes, I don't think such pretraining data can let GPT-2 understand language. If not, then the setting is not reasonable, and facts may have been learned in GPT-2 before training.

In summary, I think this paper explores an interesting topic. However, the above concerns are essential to prove that the findings are correct. If the authors can address my concerns successfully, I'll be glad to raise my score to strong accept.

**Questions:**

1. Why use LLaMA-30B, not 65B or 7B?
2. For the GPT-2 small, do you "pretrain" it from scratch or pretrained GPT-2?
3. Why use LoRA for the finetuning? If I understand correctly, the model is quite small, and it's totally unnecessary to use parameter-efficient tuning. It not only reduces the test accuracy but also makes the setting harder to follow. And thus much information in Figure 2 is redundant.
4. The augmentations in section 4.2 are confusing. Why this kind of augmentation? Do you follow existing work or propose it at first? From my view, I don't think this can work as augmentation. It can just alleviate the problem of the problematic templates.
5. Why call the dataset a semi-synthetic biography? Generated from LLM doesn't mean the fact is true. I think it's still synthetic data.
6. What does the "(Part A)" mean in the title?

---

### Official Review · Reviewer_kyUn · 2023-11-01

**Soundness:** 3 good
**Presentation:** 3 good
**Contribution:** 3 good
**Rating:** 8
**Confidence:** 4

**Summary:**

This paper analyzes the gap between knowledge memorization and utilization of LLMs. Upon a carefully created benchmark, it examined the effect of different training processes and data augmentation.

**Strengths:**

* This paper provides valuable insights into knowledge memorization and extraction of LLMs, especially the influence of training processes. It could have broad impact on related research areas, such as addressing knowledge conflicts.

**Weaknesses:**

* The limited scale of the experiments restricts the generalizability of the conclusions. This paper conducts experiments on a single dataset focused on individuals, employing a relatively small model size.

**Questions:**

* Result figures are a bit hard to read due to the small font size.

---

### Official Review · Reviewer_DDxF · 2023-11-01

**Soundness:** 2 fair
**Presentation:** 2 fair
**Contribution:** 3 good
**Rating:** 5
**Confidence:** 4

**Summary:**

This paper primarily addresses the research question: How do language models memorize factual knowledge during pre-training, and how do they extract and apply this learned knowledge for reasoning and answering questions? To answer this question, the authors create two biographical text datasets, one generated using templates and the other through prompting the LLAMA model. Additionally, a question-answering dataset is developed to evaluate the model's capability to extract knowledge. The key findings through controlled experiments include:
1. During pre-training, models tend to focus on quickly learning to answer questions rather than comprehending the complete knowledge behind the answer. This behavior might hurt the model's generalization performance on unseen data.
2. If the representation in the pre-training dataset is too uniform, it might impact the model's knowledge acquisition, potentially affecting its generalizability. Hence, data augmentation to diversify knowledge representation can help the model better link related concepts, enhancing knowledge extraction.

**Strengths:**

1. The paper is well-motivated, presenting a clear research question. The authors offer a lucid explanation of the significance and rationale behind their study.
2. Utilizing synthetic datasets as a means to control variables is an ingenious approach.
3. The experimental design is also ingenious, unveiling intriguing findings about model knowledge acquisition and extraction. Notably, the discovery that models prefer learning a simplified form of knowledge extraction before grasping the related underlying knowledge during pre-training offers valuable insights into the learning process. Additionally, the probing experiment results are compelling, suggesting that if training data representation is too uniform, the model might struggle to select the right tokens for knowledge storage.

**Weaknesses:**

1. While the paper presents meaningful insights into how models learn and store knowledge, the exploration into knowledge extraction seems lacking. The findings, especially regarding data augmentation enhancing generalization, aren't particularly novel. The paper doesn't delve deeply into how models select, extract, and utilize knowledge during inference time.
2. The study appears limited in terms of tasks and datasets. It's unclear if the findings are generalizable to other domains or tasks. Exploring multiple tasks and datasets could have further validated the generalizability of the conclusions.
3. Some of the results are confusing, and locating corresponding values within the paper can be challenging. For instance, section 4.1 references pre-train + QA fine-tuning results in Appendix E, but Appendix E does not include these results.

**Questions:**

Why focus solely on biographical data? What makes you believe that results from this type of data can be generalized to other domains and tasks, thereby addressing the overarching research question about knowledge learning and extraction in language models?

---

> ### Comment · Reviewer_DDxF · 2023-11-23
>
> Dear AC and Reviewers,
>
> Because the authors did not submit a response, my concerns have not been alleviated. I will keep my scores unchanged.
>
> Best,
> Reviewer DDxF

---

### Official Review · Reviewer_vHD7 · 2023-11-02

**Soundness:** 3 good
**Presentation:** 3 good
**Contribution:** 4 excellent
**Rating:** 6
**Confidence:** 3

**Summary:**

The proposed paper, entitled "Knowledge Storage and Extraction in Language Models (Part A)," aims to explore how transformer-based language models store and extract knowledge during training and inference. The authors focus on factual knowledge, such as that found in a knowledge graph, which the language model needs to memorize from the training corpus, encode in its weights, and later extract to answer questions or perform logical reasoning during inference.

The paper addresses the following key questions:

1) How do language models memorize knowledge during training, and extract it later to answer questions or perform logical reasoning during inference?

2) Can language models be fine-tuned to extract knowledge from specific domains, and if so, how do they achieve this?

3) What is the relationship between the model's knowledge extraction ability and different diversity measures of the training data?

4) How does the model encode knowledge attributes at the hidden embedding of entity names, and is there a correlation between this encoding and the model's knowledge extraction ability?

To study these questions, the authors propose conducting an in-depth analysis using a controlled set of semi-synthetic biography data. They construct a synthetic dataset of 100k biographies, including attributes such as birthday, birth city, and major of study, and pretrain the language model on this dataset. The authors then evaluate the model's knowledge extraction ability by fine-tuning it on question and answer (QA) pairs for a fraction of individuals and testing its ability to answer QAs about the remaining fraction.

**Strengths:**

1) The authors conducted experiments to evaluate the effectiveness of knowledge augmentation and the impact of partially augmenting data on knowledge extraction for non-augmented data. The results of these experiments highlighted the significant role of these methods in improving performance on downstream tasks.

2) The paper presents a well-balanced tradeoff between knowledge control and naturalness of data by accurately generating and subsequently rewriting it using LM. This approach enhances the clarity of results.

**Weaknesses:**

1) The authors did not provide sufficient information on QA evaluation and there is no established method to extract and assess answers from generative models. The experiments were conducted solely with the GPT-2 model, which is a decoding-only transformer. It would have been beneficial to include experiment results with encoder-decoder models such as T5 or similar alternatives.

**Questions:**

1) How many tokens was used for the training model (Size of generated datasets in terms of tokens with and without augmentation)?

---

### Official Review · Reviewer_xjTh · 2023-11-02

**Soundness:** 4 excellent
**Presentation:** 4 excellent
**Contribution:** 4 excellent
**Rating:** 8
**Confidence:** 4

**Summary:**

This paper investigates a very important question regarding how LLMs extract knowledge from text. This is different from the LLMs ability to regurgitate text that it had seen during training by being able to complete generations given a prefix. Instead, this paper investigates whether LLMs are actually able to understand facts by evaluating whether it can answer questions related to what it has been trained on. However, it is hard to study this phenomenon on LLMs that have been trained on web-text because it is very hard to ensure what the LLM might have seen during training. Instead, the authors study this in a semi-synthetic setting where the LLM is trained on synthetic biographies generated by templatized texts or LLama-30B. The LLM that has been pre-trained on scratch is a GPT2-small model.

The paper considers two different training paradigms - (a) Mixed training: in which the LLM is pre-trained on all the biographies as well as 50% of the question answer pairs, and (b) Pre-train + Fine-tune setting where the LLM is first pre-trained on biographies and then fine-tuned on 50% of the QA pairs. Note during mixed-training, the LLM observes biographies and QA pairs interspersed, but in pre-training+fine-tune the LLMs first observes only biographies and then only QA pairs. The performance of the model is tested on held-out QA pairs $P_{\mathrm{test}}$

In the mixed-training setting, the LLMs are able to generalize to held-out questions but the paper shows that the learning behavior of the models are a bit abnormal. For example, instead of first learning from biographies, the LLMs instead first learns from the QA pairs (as demonstrated by rapid increase in accuracy of token prediction) and then aligns the encoded knowledge with the BIO data to learn and extract knowledge and generalize it to the held-out questions. The authors point out this is akin to “students studying to pass the test”.

The previous hypothesis is confirmed in the pre-train+fune-tune setting where the model fails to generalize at all on $P_{\mathrm{test}}$. This is surprising as in both cases the model is trained on the same amount of data. This demonstrates that LLMs are great at memorizing and regurgitating text but that is not equivalent to extracting knowledge from it.

The paper next shows that knowledge augmentation-i.e. Augmenting the BIO set with multiple paraphrases, sentence permutations, and replacing pronouns with full-names increases the generalization accuracy (9.7% -> 96.6%). A possible explanation of this hypothesis is that exposing the model to varied expression of identical knowledge lets it focus on semantics rather than surface forms. This hypothesis is confirmed by training two kinds of linear probes that indeed show that upon knowledge augmentation, the representation of entity names capture attributes about the entities that are amenable to be extracted via fine-tuning for QA. The paper also shows that this generalization also holds true when a part of the data is repeated, the model can learn and apply its knowledge to new entities.

**Strengths:**

**Originality**

* The paper is original and does an investigation of a crucial problem about how knowledge is extracted by LLMs during pre-training. To do so, the paper adopts a synthetic setting and does exhaustive experiments to prove its hypothesis

**Quality**

* The paper is of high quality. The experiments carried out to address the research inquiries are comprehensive, and the derived findings have been meticulously analyzed and interpreted.

**Clarity**

* The paper is very clearly written and explained.  It is easily accessible to readers, ensuring that they encounter no difficulties in understanding its content.

**Significance**

* The paper makes substantial contributions by addressing the vital issue of how Language Models extract information from text. It reveals that Language Models acquire knowledge in a manner that significantly deviates from human learning processes. Furthermore, the paper underscores the importance of redundancy and repetition in knowledge extraction for Language Models. This insight offers a potential explanation for the remarkable generalization abilities observed in Language Models trained on web-text, which frequently exhibits these characteristics.

* These results have several practical applications too - e.g. it might be important to rewrite infrequently occurring data inorder for LLMs to capture them during pre-training.

**Weaknesses:**

* Even though I really liked the results in the section “Celebrity might help minority”, I think an open question that is not addressed in the paper is how many celebrities are needed to save minorities. In other words, how many entities with frequently repeated information are required for generalization to unseen/new entities?
* How much effect does scale have? The current experiments have been conducted by a GPT-2 small/medium model. Is it worth considering that some of the current findings might not hold true for models which are larger by orders-of-magnitudes? Would they demonstrate different kinds of learning?

**Questions:**

* I think it would be nice to have some results/discussion regarding if all the results would hold true for truly large LMs (>= 65B parameters). It is unclear to me if the emergent capabilities of the models allow them to learn differently.
* An analysis of how much rewriting is required would also be an interesting result and could have several practical implications.

---

### Comment · Area_Chair_Lh4r · 2023-11-22

Dear authors and reviewers,

This a reminder that deadline of author/reviewer discussion is AOE Nov 22nd (today). Please engage in the discussion and make potential adjustments.

Thank you!
AC